# The complete structure of an activated open sodium channel

Altin Sula[1],[*], Jennifer Booker[1],[*], Leo C.T. Ng[2], Claire E. Naylor[1], Paul G. DeCaen[2] & B.A. Wallace[1]

Voltage-gated sodium channels (Navs) play essential roles in excitable tissues, with their activation and opening resulting in the initial phase of the action potential. The cycling of Navs through open, closed and inactivated states, and their closely choreographed relationships with the activities of other ion channels lead to exquisite control of intracellular ion concentrations in both prokaryotes and eukaryotes. Here we present the 2.45 Å resolution crystal structure of the complete NavMs prokaryotic sodium channel in a fully open conformation. A canonical activated conformation of the voltage sensor S4 helix, an open selectivity filter leading to an open activation gate at the intracellular membrane surface and the intracellular C-terminal domain are visible in the structure. It includes a heretofore unseen interaction motif between W77 of S3, the S4–S5 interdomain linker, and the C-terminus, which is associated with regulation of opening and closing of the intracellular gate.

[1] Institute of Structural and Molecular Biology, Birkbeck College, University of London, Malet Street, London WC1E 7HX, UK. [2] Department of Pharmacology, Feinberg School of Medicine, Northwestern University, 320 E Superior, Chicago, Illinois 60611, USA. * These authors contributed equally to this work. Correspondence and requests for materials should be addressed to P.G.D. (email: paul.decaen@northwestern.edu) or to B.A.W. (email: b.wallace@mail.cryst.bbk.ac.uk).

The transmembrane conduction pathway in an eukaryotic sodium channel (Nav) is formed by a pseudotetrameric alpha-subunit comprised of a single polypeptide chain with four repeating homologous regions. Prokaryotic sodium channels, in contrast, are true tetrameric structures, composed of four identical subunits, each one equivalent to a single eukaryotic region. Both eukaryotic and prokaryotic Navs consist of voltage sensor (VS) and pore domains, which are responsible for regulation and ion-translocation functions, respectively, and C-terminal domains (CTDs), which appear in both cases to have regulatory functions, albeit of quite different natures and structures. Prokaryotic orthologues exhibit approximately 25–30% sequence identity with human Navs and several have been shown to have sodium ion translocation activity[1–4] and one (NavMs from *Magnetococcus marinus*) has been shown to be blocked by human sodium channel blockers with similar affinities and kinetics as the human Nav1.1 isoform[3], thus indicating the structure/function similarities of eukaryotic and prokaryotic Nav orthologues.

Whilst no structures have yet been produced for eukaryotic sodium channels, a number of crystal structures of prokaryotic sodium channels in different conformations with different regions of their structures visible[1–3,5–9] have provided insights[10] into functionally-related structural features of this important family of ion channels. None of the prokaryotic sodium channel structures solved to date consist of all three domains, voltage sensor, pore and C-terminal domains.

In this study, the high-resolution crystal structure of a complete voltage-gated sodium channel (with crystals formed under zero potential conditions consistent with an open-channel state) has been determined. The structure displays an activated VS domain conformation, three sodium ions present in its open selectivity filter, and an open intracellular pore gate formed from the S6 transmembrane helices leading into a regulatory CTD. It has revealed the nature of an interaction domain that forms the basis for understanding the opening and closing of the intracellular gate, and indicated a set of residues essential for enabling the open conformation, including signature residues that are conserved across prokaryotic and eukaryotic sodium channels. Functional studies modifying those residues have demonstrated their essential roles in the cycle of opening/closing/inactivation. Hence, the present study combining structural biology and electrophysiology has enabled identification of the molecular basis of gating of this sodium channel.

## Results

**The overall structure**. This new crystal structure (Fig. 1, Table 1, Supplementary Figs 1 and 2) of the NavMs channel is the highest-resolution structure yet determined of a sodium channel and

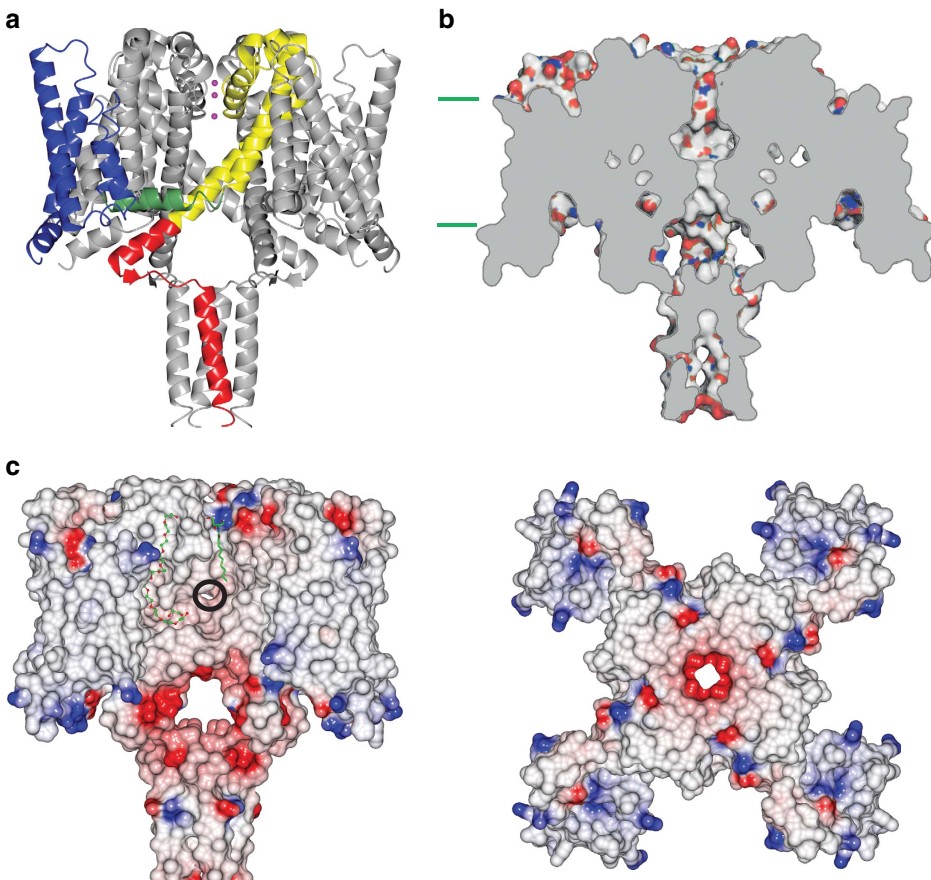

**Figure 1 | Structure of the NavMs channel.** (**a**) 'Cartoon' representation of the NavMs tetrameric channel. One of the monomers is coloured as follows: voltage sensor (blue), S4–S5 linker (green), pore helices (yellow), C-terminal domain (red), and sodium ions (purple). The other three monomers are depicted in grey, for ease of viewing. (**b**) Cross-sectional view, showing the open SF, hydrophobic cavity, and open pore gate. The opening does not extend into the stabilising coiled-coil bundle at the end of the CTD, but provides an egress via the bundle linker region. The approximate locations of the membrane surfaces are indicated by the cyan lines. (**c**) Space filling model (coloured according to electrostatics). The locations of detergent and polyethylene glycol molecules are shown on the surface in green and red stick format. Left: view from the membrane normal. The location of the fenestration leading to the hydrophobic cavity is circled in black. Right: view from the extracellular surface, extending from the N-termini to the ends of the S6 helices.

## Table 1 | Data collection and refinement statistics (molecular replacement).

| | NavMs wild type (5HVX) | NavMs I218C (5HVD) |
|---|---|---|
| *Data collection* | | |
| Space group | I422 | I422 |
| Cell dimensions | | |
| *a, b, c* (Å) | 109.0, 109.0, 210.6 | 109.6 109.6 209.7 |
| *α, β, γ* (°) | 90.0, 90.0, 90.0 | 90.0 90.0 90.0 |
| Resolution (Å) | 48.4–2.4 | 58.6–2.6 |
| | (2.55–2.45)* | (2.72–2.60) |
| $R_{merge}$ | 0.30 (2.58) | 0.29 (3.60) |
| $R_{pim}$ | 0.06 (0.49) | 0.045 (0.095) |
| $I/\sigma(I)$ | 16.5 (1.4) | 15.8 (1.4) |
| $CC_{1/2}$ | 1.00 (0.77) | 0.99 (0.70) |
| Completeness (%) | 100.0 (100.0) | 100.0 (99.7) |
| Redundancy | 28.2 (29.3) | 43.3 (45.5) |
| | | |
| *Refinement* | | |
| Resolution (Å) | 27.7–2.45 | 26.57–2.60 |
| No. of reflections | 670,458 | 868,621 |
| $R_{work}/R_{free}$ | 0.209/0.238 | 0.200/0.228 |
| No. of atoms | | |
| Protein | 1,980 | 1,976 |
| Ligand/ion | 85 | 69 |
| Water | 100 | 111 |
| *B* factors | | |
| Protein | 98.9 | 93.4 |
| Ligand/ion | 120.8 | 104.0 |
| Water | 93.9 | 86.0 |
| R.m.s. deviations | | |
| Bond lengths (Å) | 0.010 | 0.010 |
| Bond angles (°) | 1.02 | 1.05 |

5HVX refinement based on a single data set taken from a single crystal, 5HVD refinement based on two data sets collected from a single crystal.
$R_{merge} = \Sigma(I - <I>)/\Sigma<I>$
$R_{work} = \Sigma(|F_{obs}| - |F_{calc}|)/\Sigma|F_{obs}|$ for 95% of the data. $R_{free}$ is the same definition but for the 5% of the data excluded from refinement.
*Values in parentheses are for highest-resolution shell.

uniquely includes the entire molecule: the N termini, VS domains (transmembrane helices S1 to S4), S4–S5 linkers, pore domains comprised of the S5 to S6 transmembrane helices (including the selectivity filter (SF) with its ions), and the CTD. This structure has activated VSs, an open pore gate, and a CTD domain that interacts with the S4–S5 linker between the pore and VS domains; hence, it represents an open, activated state of the channel. This differs from previous studies of closely related Nav orthologues, where the structures were of a 'pre-activated' state with a closed gate[1], 'potentially inactivated' forms with collapsed SFs and closed gates[5,6], a pore-only construct with an open gate[2,8], and a pore-CTD construct with a closed gate[6]. The juxtapositions of all of the elements in the present structure provide new information about structurally related functional characteristics, and enable a much more complete understanding of the functional linkage between the structural elements[10].

**An activated voltage sensor domain.** The VS is in the fully 'activated' state, with the outward gating charge position exposed to the extracellular surface (Fig. 2a), and indeed, the activated state seen here is what would be expected at the 0 mV condition present in the crystal. The channel activation process has been proposed to arise from the transmembrane movement of charges[4], produced by translocation of the S4 helix relative to the other helices in the VS, forming alternative ion paired or hydrogen-bonded sets between four arginines (referred to as

R1–R4) in S4 and adjacent negatively charged (D or E) residues in S1, S2 and S3 (Fig. 2d). Previous crosslinking studies[11] on the NaChBac orthologue from *Bacillus halodurans* (Fig. 2d), indicated that the activated state associated with the outward facing positions would enable ion pairings involving all four arginines with the C-terminal-most arginine (R4; NavMs residue 106) pairing with the equivalent to NavMs residues E59 in S2 and D81 in S3. This set of ion pairs is seen in the present crystal structure and also in the NavAb structure (Fig. 2a,b). The pairing of NavMs R2 and R3 with the side chain of E46, D49, and N25 in helices S2 and S1, respectively, are also clear (Fig. 2a). R1 is located at the extracellular end of S4 and is not ordered in the crystal structure. The NavAb orthologue sequence, in contrast, (Fig. 2d) has no negatively charged side chains equivalent to those of the E46 or D49 residues, which are involved in the canonical 'activated' pairings in NavMs. In the NavAb structure[1] (which has been described as being in a 'pre-activated' state), R4 forms ion pairs with the same residues (E59 and D80) as in NavMs (Fig. 2b), but in the NavMs structure the R4 site includes an additional ion pair and altered conformation from that in the NavAb pre-activated state. In the NavAb structure, R2 does not form an ion pair with any residue side chain in the S2 or S3 helices, just a hydrogen bond with the main chain carbonyl of V89 in the S3 helix, and R3 only forms an ion pair with N25 in S1, whereas in the present structure R3 also pairs with D49. Hence, this new structure is proposed to represent the 'activated' state, in comparison with the 'pre-activated' state seen in NavAb. The root mean squared deviation (r.m.s.d.) of 1.6 Å between the NavMs and NavAb S4 structures (backbone atoms and the R2, R3 and R4 side chains) further reflects this substantial difference. In a NavAb-humanNav1.7DIV chimeric structure[9], the side-chain pairings are more complete and more similar to those in our structure, and the r.m.s.d. between it and NavMs is somewhat closer at 1.3 Å. For comparison, a recent structure of the related two-pore channel (TPC1)[12] has its VS in the fully inwardly facing 'resting state' position[11] (Fig. 2c,d), as it uses an entirely different set of pairings for R1–R4.

**An open pore.** The NavMs pore gate in this structure is in an open conformation. The pore domains in all the Nav structures consist of a funnel-like vestibule on the extracellular membrane surface leading into the SF, which is a narrow restriction that provides the selectivity for sodium versus other ions (Fig. 1a,b). The conformations and dimensions of the 'open' SFs present in the open NavMs structure (this work) and the pre-open NavAb structure[1] are nearly identical (r.m.s.d. ~0.4 Å) (Fig. 3a), and both are of sufficient size to enable sodium ions to pass, although sodium ions are only seen in the SF of the NavMs structure. The three SF sodium ions and waters visible in the SF of this open-channel structure are virtually indistinguishable from those previously identified in the structure of the open pore-only NavMs construct[13] (Supplementary Figs 1,3). The SFs in the 'inactivated' crystal structures of NavAb[5] and NavRh[6] are collapsed and hence would not allow transit of sodium (or other) ions. The SF in NavMs leads to a large hydrophobic cavity in the transmembrane region (Figs 1b and 3a). All of the Nav structures, whether open or closed, have similar hydrophobic cavities (where sodium channel blocker molecules have been shown to bind[3]) (Fig. 3a). The cavity leads to a narrowing (gate) at the intracellular end of the pore, formed by the conjunction of the S6 helices. When this gate is in the closed state (as it is in all structures determined to date except the NavMs pore only[2,3,8] and this full-length NavMs channel structure), it is too narrow to allow the passage of sodium ions (Supplementary Fig. 4). However, in the open state, as in both NavMs structures, ions

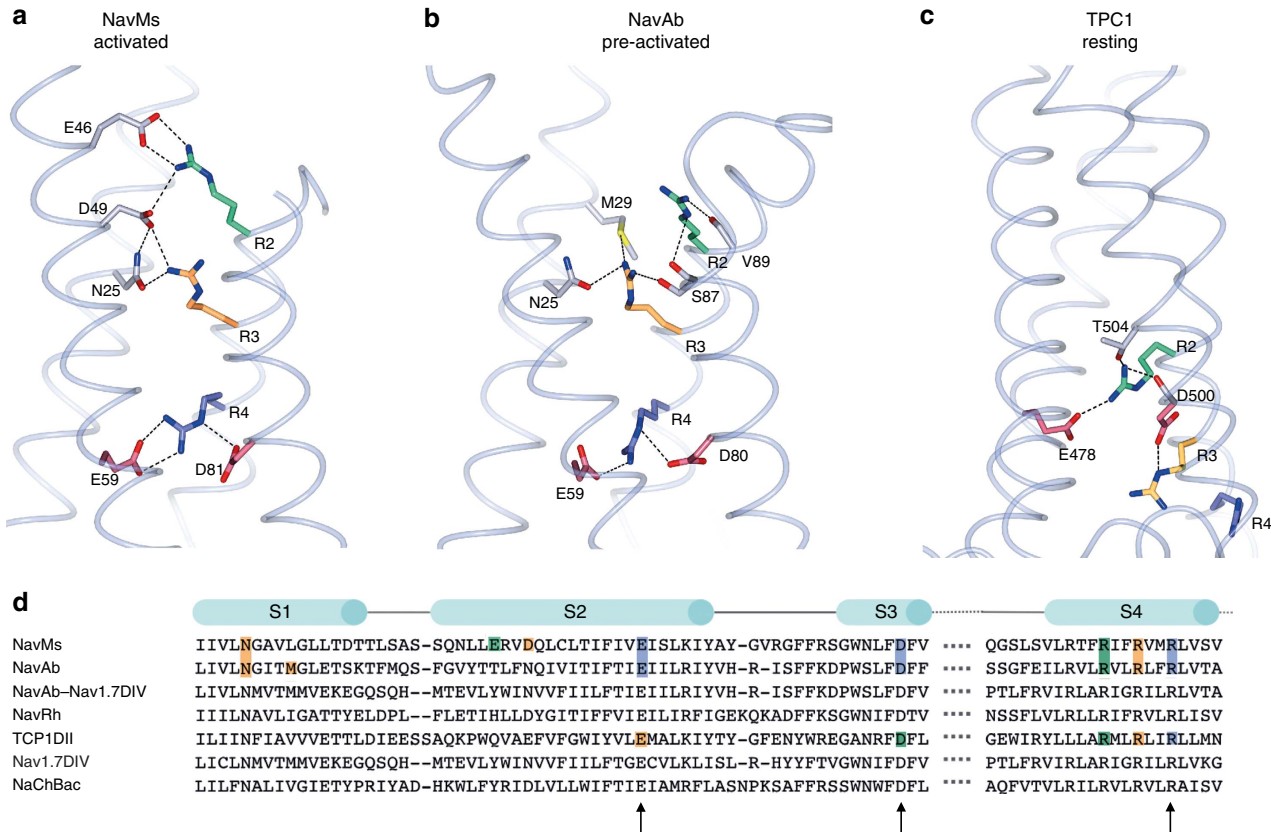

**Figure 2 | Comparisons of voltage sensors.** Ribbon representations of (**a**) NavMs activated, (**b**) NavAb pre-activated and (**c**) TPC1 resting state channels, including side chains of the residues in S1, S2 and S3 which form ion pairs with the S4 arginine residues. (**d**) Sequence alignments of the NavMs, NavAb, NavAb-Nav1.7DIV chimera, NavRh, TPC1 domain II, human Nav1.7DIV and NaChBac VS sequences (colour-coded by pairings as in **a**–**c**) with their corresponding partners. Crosslinked positions in NaChBac from ref. 11 are annotated by arrows below the sequences.

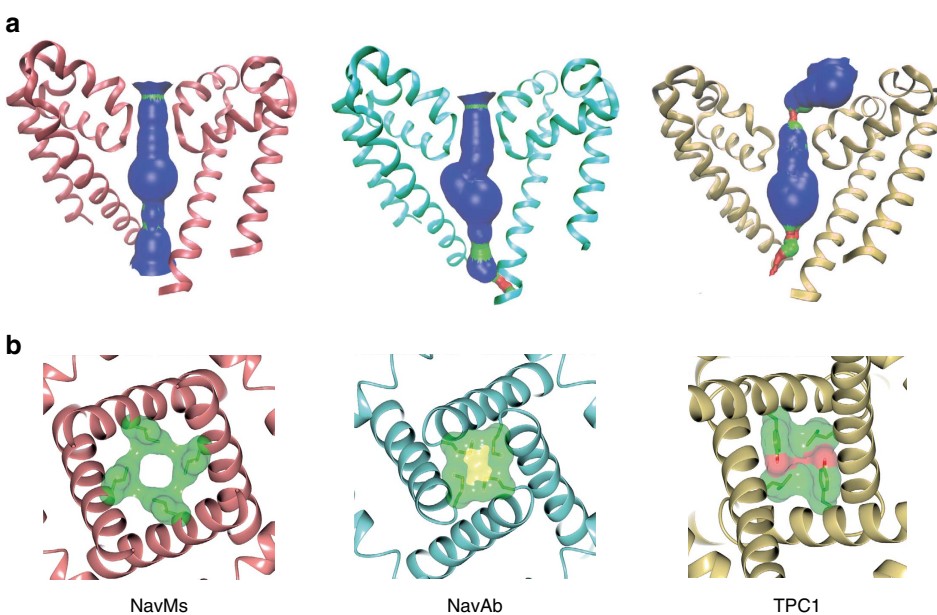

**Figure 3 | Comparisons of pore sizes and gates.** (**a**) Plots showing the accessible interior of the pore domain and size of the pore gate calculated with HOLE[30] for NavMs (open pore), NavAb (closed pore) and TPC1 (closed pore), where blue surfaces (pore radius >2.3 Å) are compatible with sodium ion translocation, green surfaces are radii between 1.2 and 2.3 Å, and red <1.2 Å. (**b**) Views (from the intracellular side) of the pore gates for NavMs, NavAb and TPC1 channels overlaid with semi-transparent space-filling models of the gate residues (point of narrowest constriction), coloured by the atom type (NavMs: Ile215; NavAb: Met221; TPC1: Tyr305 and Leu672).

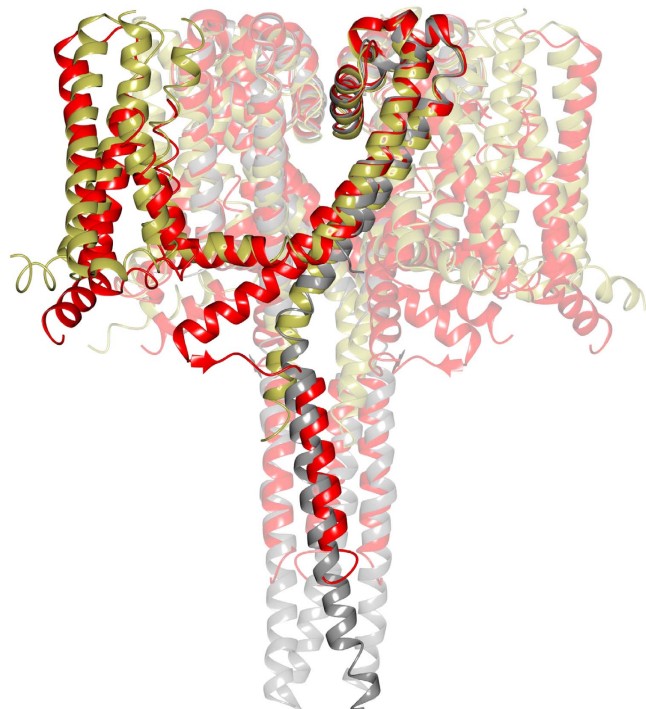

**Figure 4 | Structural alignments comparing the CTDs of open and closed sodium channels.** Alignments of Nav structures that have all (NavMs (red) and NavAe1 (grey)) or part (NavAb-humanNav1.7DIV chimera (gold)) of their C-terminal domains visible in crystals, showing the different orientations and structures at the end of the S6 pores. Only the open pore NavMs structure shows the stabilising interactions with the S4–S5 linker. The overlap of the C-terminal coiled-coils is evident in NavMs and NavAe1 where the distal ends are visible. These structures are compatible with the opening/closing models previously suggested[2,7,8].

can pass unimpeded. In the NavMs channel, the pore opening is slightly narrower than in the NavMs pore (r.m.s.d. of the S6 alpha carbons ∼0.4 Å) (Supplementary Fig. 4), but this is still more than sufficient for the passage of at least partially-hydrated sodium ions (Fig. 3a,b), and much larger than the openings in any of the closed structures. The differences between the intracellular ends of the transmembrane parts of the S6 helices in the NavMs pore and channel structures may arise from constraints imposed by the presence of the ordered, open CTD domain in the full-length channel structure (Figs 1 and 4). The conformation of the CTD in the NavAe1 structure[7], where the pore is in a closed state, is very different (Fig. 4).

The activated S4–S5 linker enables opening of the gate. It was anticipated that the S4–S5 linker (Figs 1 and 5) could act as a lever between the VS and pore domains, opening the pore gate produced by the conjuncture of the S6 helices. Whilst the S4–S5 linker was not present in the pore-only NavMs structure, it was suggested[2] that if that structure had retained the relative orientations found in the pre-activated NavAb structure, there would have been a clash between the VS conformation and the adjacent S5 in a full-length open structure. This conjecture was confirmed by the new structure. Superposition of the pre-activated NavAb VS onto that of the activated NavMs VS (Fig. 5a; Supplementary Fig. 5), produces a large shift in the C-terminal end of the linker, where it meets the S5 helix of the pore domain, suggesting that activation of the VS enables the linker to act as a lever which, in turn, shifts the positions of the pore domain helices S5 and S6, thereby opening the gate (Fig. 5b).

**The C-terminal domain and interaction motif.** The CTD of an open-state sodium channel is uniquely visible in this structure. It begins in the C-terminal region of the S6 helix, which extends beyond the transmembrane region of S6 (that ends at residue M222) through to the end of this helical region (residue K233). It then continues, first, into a short non-helical region, and then into the intersubunit four-stranded coiled-coil region (Fig. 5a). This is very different than the essentially fully helical CTD structure of the closed NavAe1 pore[13] (Fig. 4).

The extension of the S6 helix (which is the proximal region of the CTD closest to the gate) forms an interaction motif involving specific salt bridges and hydrogen-bonded interactions with the S4–S5 linker adjacent to the intracellular surface and with Trp77 in S3 (Fig. 5a). In the EEE motif (residues 229–231) near the end of the extended S6 helix, E229 forms a salt bridge with R119 in the S4–S5 linker, whilst E230 and E231 form hydrogen bonds with N238 and H237, respectively, in the top of the 'random coil' region, leading to the C-terminal coiled-coil. R118 and Q122 in the S4–S5 linker interact (by aliphatic stacking and a hydrogen bond, respectively) with Trp77 in S3, a residue that is fully conserved among prokaryotic and human sodium channels (all isoforms and all domains) (Fig. 5c; Supplementary Fig. 6) but which is not conserved in other ion channels such as potassium and TPC channels (Supplementary Fig. 7). This interaction motif creates a communication network between the linker, S3, and S6. Along with the Trp in S3, the cluster of negatively charged residues at the beginning of the C-terminal region (Fig. 5c) also appears to be a common feature in prokaryotic and eukaryotic sodium channels.

Farther along the CTD, a 'random coil' structure leads into the coiled-coiled bundle, which begins around residue 239 and extends to the C-terminal end (Fig. 5a; Supplementary Fig. 3). It is the conformational change between the random coil structure (as seen in the open NavMs structure) and the fully helical structure (as seen in the closed NavAe1 structure[7]) that would enable a change in dimensions of the pore gate during opening and closing without dissociating the tetramers, in agreement with previous proposals[2,7,8]. Beyond residue 239 in NavMs is the four helix coiled-coil, which is the region of association between the monomers in the tetramer.

**Relationship of structure to functional properties.** Previous electrophysiological studies[8] demonstrated that removal of the entire CTD domain of NavMs, or replacement of residues 229–231 (sequence EEE) with a QQQ sequence impaired the rate of slow inactivation without significantly modifying inactivation recovery, whilst having little effect on the voltage dependence of activation and inactivation. Those results can now be rationalized from the present structure: the deletion of the upper (proximal) part of the CTD or the replacement of the glutamates with glutamines would remove the restraining interaction motif, and hence produce the observed effect on gating and inactivation. However, in the previous studies, removal of the distal region (beyond residue 239) did not affect the inactivation[8], whilst it did reduce the current density. This too can be explained by the present structure: removing the inter-tetramer coiled-coil structure would potentially lead to less stable open tetramers, resulting in fewer complete channels on the plasma membrane.

To examine the role of the conserved Trp77 in NavMs function, in the present study this residue was mutated to Ala, Met, Phe and Tyr, and sodium currents were recorded from HEK293T cells overexpressing mutant channels (Fig. 6a). Cells transfected with either the W77A or W77M mutants failed to produce any currents above background levels (Fig. 6b), whereas the W77F and W77Y mutants both conducted voltage-dependent

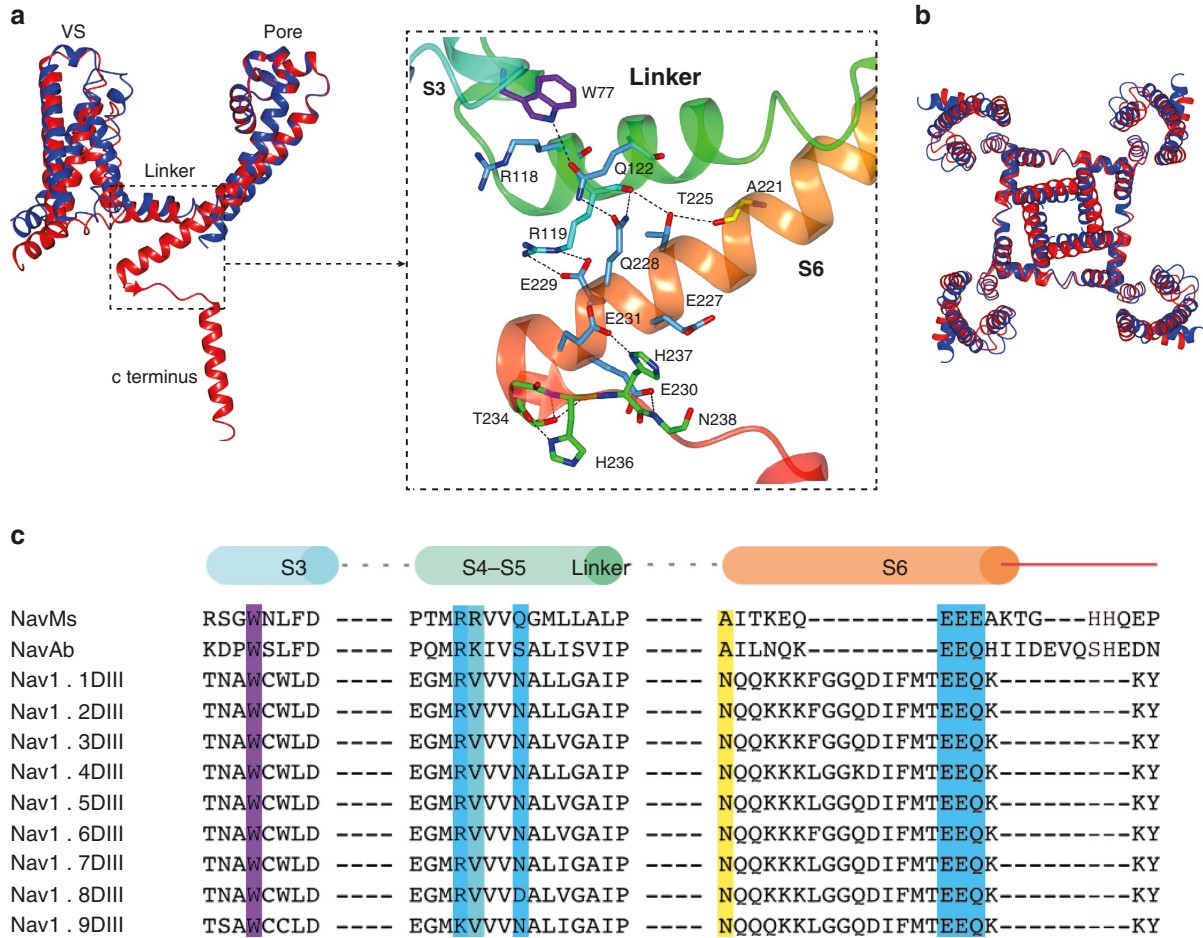

**Figure 5 | The S4–S5 linker and interaction motif.** (**a**) Overlay of the VS regions of NavMs (red) and NavAb (blue), showing the resulting displacement of the S4–S5 linkers that leads to changes in the juxtapositions of the S5 and S6 helices, producing a wider (open) pore gate in NavMs. The expanded view is of the interaction motif: the region of S3 where W77 is located (ribbon motif, in cyan), the S4–S5 linker (in green), the end of the S6 helix (in orange), and the residues at the top of the NavMs CTD (in blood orange). The colour coding is the same as in the cylinders at the top of **c**. The residues that form ion pairs and hydrogen bonds (shown in stick representation) are colour-coded as in the sequence overlays in **c**. (**b**) Superposition (as **a**) of the NavMs (red) and NavAb (blue) tetramers, viewed from the extracellular surface, showing that the shift in the linker leading to helices S5 and S6 produces a wider pore gate in the NavMs activated open structure. (**c**) Sequence alignments of S3, S4–S5 linker and S6-CTD regions of NavMs, NavAb and domains III of human Navs, highlighting the interacting residues in NavMs and the corresponding residues present in the orthologues. They are colour-coded (overlaid) as follows: the conserved Trp in S3 (purple), the residue that forms the hydrogen bond involving polypeptide backbone (yellow), the residues involving side chains (blue), and those involving both types of interactions (cyan). The cylinders at the top are colour-coded as in **a** (right) to indicate where these residues are in the NavMs structure.

sodium currents with an attenuation of the current density when compared with wild-type channels (Fig. 6b), consistent with the proposed role of the Trp in stabilising an open gate. It is unclear whether the Ala and Met mutations prohibit channel function or simply impair trafficking to plasma membrane, but the less 'drastic' Tyr and Phe substitutions clearly alter channel gating.

In addition to this effect on gating, compared with wild-type NavMs, the voltage-dependent activations of W77F and W77Y channels are shifted by 17 and 13 mV, respectively (Fig. 6a), whilst reducing the slope of the conductance-voltage relationship by 0.2–0.3 × (Table 1, charge value). W77Y and W77F mutants also exhibit 15–22 mV positively-shifted voltage dependences of inactivation, which can be attributed to their shifted voltage dependence of activation (Fig. 6a,c). These results indicate that Tyr and Phe substitutions in S3 reduce the energy required to activate the channel by $\approx 3 \, \text{kcal} \, \text{mol}^{-1}$ (2.8 and 3.1 kcal mol$^{-1}$, respectively) and reduce the kinetics of the maximum speed of activation and inactivation by 1.7–2x (Fig. 6c; Supplementary Table 1), but have no effect on the rate of recovery from

inactivation (Fig. 6d). Taken together, these data demonstrate that the W77 site contributes both to the steep and negatively shifted voltage dependence of NavMs channel activation/inactivation and gating, as expected, since W77 is located in S3 of the VS and is part of the lower gate interaction motif.

In the open NavMs structure Trp77 forms both a non-polar stacking and a hydrogen-bonded interaction with Q122. When the hydrogen-bonding potential is removed in the W77F mutant, it still retains an aromatic ring structure and the potential for a non-polar interaction with Q122 (Supplementary Fig. 8). This may still stabilize the open state (albeit less effectively) and hence explain the lower current density observed with this mutation. The W77Y mutant retains both the aromatic ring and potential for hydrogen-bonding (with some conformational adjustment) with Q122, which could account for its less diminished current density. The smaller W77A and W77M mutants would not retain either of the potential interactions; this may or may not be an explanation for its observed lack of conductance. Nevertheless, it is clear from these mutational/functional studies that the

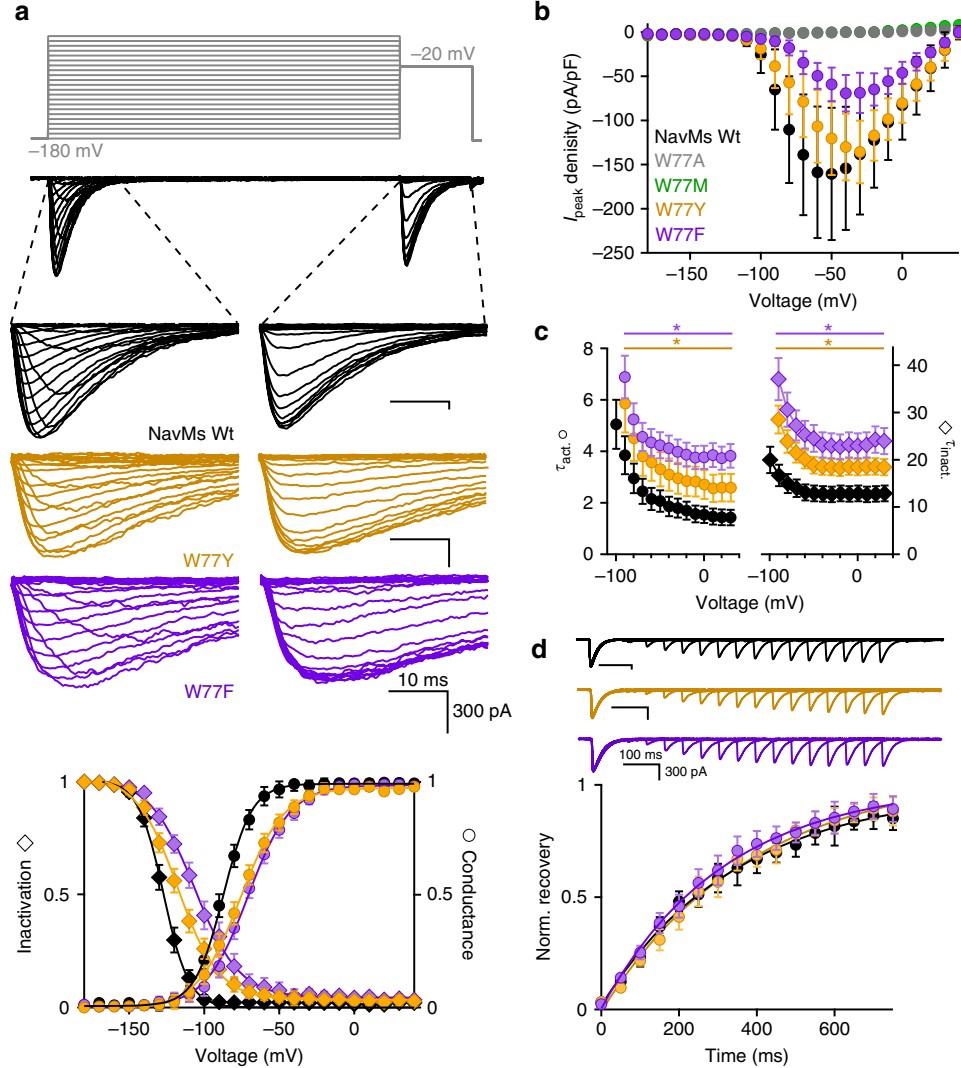

**Figure 6 | Perturbations of NavMs channel kinetics and voltage dependence by W77 mutations.** (**a**) The voltage dependence of NavMs channels. Top, grey lines indicate voltage protocol use to open and inactivate the sodium channels. Middle, exemplar sodium currents conducted by the indicated wild type (WT) and mutant NavMs channels with expanded time scale indicated by the dashed line. Bottom, resulting voltage dependence of normalized channel conductance and inactivation. (**b**) Current density of Wt and W77 mutant NavMs channels (**c**) Average time constants of entry into activation ($\tau_{act.}$) and inactivation ($\tau_{inact.}$) obtained by fitting the initial rise and subsequent decay of the sodium currents (see Methods) to a single exponential. The star indicates $P < 0.05$ resulting from a two tailed Student's $t$-test compared with the Wt values. (**d**) Rate of recovery from inactivation. Top, exemplar sodium currents were completely inactivated by a 150 ms depolarization 40 mV positive to the $V_{1/2}$ of activation. The membrane potential was then held at $-180$ mV for varying time intervals; recovery was assessed by a second depolarization. The proportion of second to the first depolarization is the normalized recovery from inactivation. For all of the results in this figure, $N = 8$ cells, Error $=$ s.d.

conserved Trp in the interaction motif plays a significant role in both channel structure and function.

This interaction motif involving the CTD is thus proposed to form a physical means of stabilizing the open state and regulating the opening/closing/slow inactivation processes seen in prokaryotic channels[14]. Previously prokaryotic CTDs were only functionally associated with stabilization of the tetrameric structure[15,16] (although not essential for formation of tetramers[16]), but this study now indicates its additional role as part of the interaction motif which distinguishes the open from the closed forms of the channel.

It is important to note, however, that other than a short segment including the cluster of glutamates just preceding the terminal coiled-coil of NavMs, (Fig. 5d; Supplementary Fig. 6), the sequences of the CTDs of human sodium channels are very different from those of prokaryotic channels. There is a crystal

structure of a C-terminal fragment of the human Nav1.5 channel (residues 1773 to 1940) available[17], but it does not include any of the voltage sensor, linker or pore regions, so direct structural comparisons with the interaction motif of NavMs are not possible. The Nav1.5 fragment does include a C-terminal region that is helical, but in NavMs the C-terminal helix, is not an isolated helix but part of the intersubunit coiled-coil structure. As eukaryotic channels are monomeric and thus have only one C-terminal helix, any similarities between prokaryotic and eukaryotic Navs in the distal end of the C-terminal region are probably not indicative of common functional roles.

## Discussion

Due to their important roles in disease[18] and as targets for drugs, neurotoxins and insecticides, there is great interest in the structure and function of Navs. Eukaryotic sodium channel

structures have been elusive, in large part due to their long and pseudorepeating nature. This has led to a focus on the structures of related prokaryotic sodium channel orthologues. All of the previously determined sodium channel crystal structures have been missing one or more domains. The present structure is the only complete crystal structure of any sodium channel determined to date. It is also the only sodium channel with an activated VS and a pore gate sufficiently wide to enable the passage of sodium ions, both features expected for an 'open' state channel, and as such it is likely to form the basis for new molecular dynamics simulations of ion translocation. This structure has furthermore enabled the visualization of new interactions between all the domains in the channel, providing a clear structural basis for understanding the linked regulation of activation and channel gate opening.

In summary, the new high-resolution NavMs structure provides a number of important structural and mechanistic insights into the functioning of sodium channels: (1) the physical linkage mechanism between activation of the VS and opening of the channel pore gate; (2) the role of the CTD in maintaining an open channel (and regulating opening and closing of the gate); (3) the identification of a new interaction motif, with features common to prokaryotic and eukaryotic members of the sodium channel family; and (4) important ways in which sodium channel structures differ from other members of the ion channel family.

## Methods

**Materials.** Quick ligase, restriction enzymes and DH5a chemically competent cells were purchased from New England Biolabs. Syntheses of PCR primers and all DNA sequencing were performed by Eurofins MWG Operon. Ni-NTA and all DNA purification supplies were purchased from Qiagen, Ltd (UK). Thrombin and the pET15b vector were purchased from Novagen, Inc (EMD Chemicals, Darmstadt, Germany). Dodecyl-β-D-maltopyranoside (DDM) and decanoyl-*N*-hydroxyethylglucamide (HEGA-10) were purchased from Anatrace (Ohio, USA). Protease inhibitor cocktail tablets were purchased from Roche (Switzerland).

**Expression and crystallization.** The full-length construct of NavMs was created in a pET15b vector (with an N-terminal hexa-His-tag followed by a thrombin cleavage site) using the C-terminally truncated construct of the gene from *Magnetococcus sp.* (strain MC-1)[2], which was joined with the C-terminus from a codon-optimized synthetic gene, starting at residue H237. For the comparisons made in this study, we used the wild type NavMs (PDBID 5HVX), as that was the sequence that had been used in the previous electrophysiology studies[3]. However, we also made an I218C mutant (PDBID 5HVD) because comparisons were also made with the NavAb structure (PDBID 3RVY), which had the I217C (equivalent residue) mutated (Supplementary Fig. 6), and we wanted to ensure differences observed were not due to that mutation. The I218C mutation was introduced into NavMs using SLIM cloning. Plasmids were PCR amplified using KAPA Hifi HotStart mix (Kapa Biosystems), with forward primer 5′-ATTGGCATTTGTGT AGATGCAATGGCAATCACCAAG-3′ and reverse primer 5′-CATCTACACAA ATGCCAATAAACAGGTTGAGCACGG-3′. All plasmids were sequenced before use. Note that although the wild type and mutant NavMs structures were essentially identical, with an r.m.s.d. (C-alphas only) of 0.25 Å (Table 1, Supplementary Fig. 4) thus demonstrating the differences discussed are not due to the I218C mutation, the corresponding r.m.s.d. between the wild type and mutant NavAb structures was considerably larger, 1.22 Å (see Supplementary Fig. 4).

The proteins were overexpressed in C41(DE3) cells upon addition of 0.5 mM isopropylthiogalactoside at $A_{600}$ nm 0.8 for 3.5 h at 37 °C. Membranes were solubilised in 20 mM Tris, pH 7.5, 300 mM NaCl, 20 mM imidazole and 1% DDM. Both proteins were purified by loading onto a HisTrap HP (GE Healthcare). The column was washed with 20 mM Tris–HCl, pH 7.5, 300 mM NaCl, 20 mM imidazole and 0.52% HEGA-10, and the proteins were eluted using a 20 to 500 mM imidazole gradient. The protein was treated with thrombin protease for 16 hours at 4C, then the sample was further purified by size exclusion chromatography using a Superdex 200 increase 10/300 column (GE Healthcare). Protein was concentrated to 10 mg ml$^{-1}$ for crystallization and either used directly or stored at −80 °C.

Crystals of both wild type and the I218C mutant were grown by the sitting drop method in 0.05 M NaCl, 0.1 M MES buffer, pH 6.5, 37.8% v/v polyethylene glycol (PEG) 400 and 4% ethylene glycol. They were flash-frozen in liquid nitrogen, where the PEG 400 present in the crystallization conditions acted as a cryo-protectant.

**Crystal structure determination.** Crystals were screened and X-ray diffraction data collected at several Synchrotron radiation sources: Diamond Light Source,

beamline I24; Soleil Synchrotron, beamlines Proxima 1 and 2; and the European Synchrotron Radiation Facility, beamline ID23. The resolutions of the crystals ranged from 3.5 to 2.45 Å. Because there was substantial variation in unit cell dimensions between different crystals produced under the same conditions, as we have seen previously for the pore-only construct[13], we were unable to merge data sets between crystals but merged several sets collected from different regions of the same crystal. The structures described are therefore based on the highest-resolution crystals identified for each type (2.45 Å for the wild type; 2.60 Å for the mutant). The diffraction images were integrated and scaled using the XDS software package[19] and merged using Aimless[20] in the CCP4 program suite[21].

The structures were determined by molecular replacement using the full-length NavAb structure (PDB 3RVY) as the search model. Phases were obtained with Phaser[22], and model building was performed in Coot[23]. They were initially refined in Refmac[24], and then the refinement was continued in Buster[25]. The data collection, processing and refinement statistics are listed in Table 1. The refinement quality was checked using MolProbity[26] which indicated that 100% of residues were in allowed conformations. Figures were created in CCP4mg[27], unless otherwise noted.

**Bioinformatics.** Sequence alignments were done using the Clustal Omega multiple sequence alignment tool[28] or EMBOSS Needle pairwise alignment tool[29], with the following Uniprot codes: NavMs (A0L5S6), NavAb (A8EVM5), NavRh (D0RMU8), NavAe (Q0ABW0), NaChBac (Q9KCR8), TCP1 (Q94KI8), Nav1.1 (P35498), Nav1.2 (Q99250), Nav1.3 (Q9NY46), Nav1.4 (P35499), Nav1.5 (Q14524), Nav1.6 (Q9UQD0), Nav1.7 (Q15858), Nav1.8 (Q9Y5Y9), Nav1.9 (Q9UI33).

Structural alignments and figures were generated using CCP4mg[27]. The following PDB entries were used for structural comparisons: NavMs pore (3ZJZ), NavAb (3RVY), NavRh (4DXW), NavAb-Nav1.7VSDIV (referred to as 'chimera') (5EKO), TCP1 (5E1J), NavAe (4LTO).

Inner pore measurements were calculated using the HOLE suite of programmes[30] and visualized with the Visual Molecular Dynamics (VMD) program[31].

**Whole-cell voltage clamp electrophysiology.** HEK293T cells (acquired through ATCC(CRL-3216)) were transiently transfected with pTracer CMV2 plasmids (Supplementary Fig. 2) containing NavMs and GFP genes, and voltage clamped in the whole-cell configuration[3]. The external solution contained 150 mM NaCl, 10 mM HEPES, and 1.8 mM CaCl$_2$, adjusted to pH 7 with NaOH and to an osmolality of $300 \pm 5$ mOsm with mannitol. The internal solution contained 100 mM CsMES, 30 mM NaCl, 10 mM HEPES, 5 mM EGTA, 3 mM MgCl$_2$, and 0.5 mM CsF. CaCl$_2$ was added to achieve a final free Ca$^{2+}$ concentration of 100 nM. The pH and molarity were adjusted to 7.4 with CsOH and $300 \pm 5$ mOsm with mannitol, respectively[32]. Leak current and capacitance transients were subtracted using a standard P/4 protocol. Data from cells were disqualified if they did not meet all of the following criteria: voltage error must be $>3$ mV, expression of maximum NavMs must exceed 150 pA, and leak current at $-180$ mV must be less than $-100$ pA. Clampex 10 (Molecular Devices, Sunnyvale, CA) was used to acquire the voltage clamp data. pClamp 10 (Molecular Devices, Sunnyvale, CA) and Igor Pro 7.1 (Wavemetrics, Lake Oswego, OR) were used to analyse the data. The time constants ($\tau$) measuring activation, inactivation and the recovery from inactivation were determined by fitting the data to the exponential function: $f(x) = B + A \exp[(1/\tau)x]$, where $\tau$ is the half-time constant current decay or recovery rate, and $x$ is time. The current-voltage relationships (Fig. 6) were determined by fitting the data to the following equation: $(V - V_{Rev})/\{1 + \exp[(V - V_{1/2})/K]\}$, where $V_{Rev}$ is the extrapolated reversal potential, $V_{1/2}$ is the half-activation voltage, and is a slope factor equal to RT/ZF, Z is the apparent gating charge, and F is Faraday's number. The free energy required to shift the channel from the open to closed state was calculated as $\Delta G° \ (\text{kcal mol}^{-1}) = 23.06Z(F)V_{1/2}$. The error bars in Fig. 6 represent standard deviations of currents measured from eight cells.

**Data availability.** Structure factors and coordinates for the wild type and I218C mutant have been deposited in the Protein Data Bank with accession codes 5HVX and 5HVD, respectively. All the other data supporting the findings of this study are available from the corresponding authors, upon reasonable request.

The Uniprot accession codes A0L5S6, A8EVM5, D0RMU8, Q0ABW0, Q9KCR8, Q94KI8, P35498, Q99250, Q9NY46, P35499, Q14524, Q9UQD0, Q15858, Q9Y5Y9, and Q9UI33 and the PDB files 3RVY, 3ZJZ, 4DXW, 5EKO, 5E1J, and 4LTO were used in this study. The plasmids are available from PGD (Paul.Decaen@northwestern.edu).

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

## Acknowledgements

This work was supported by grants BB/L006790 and BB/L026251 from the U.K. Biotechnology and Biological Science Research Council (BBSRC) (to B.A.W.) and an NIH NIDDK Pathway to Independence (PI) Award (K99/R00) (to P.G.D.). J.B. was supported by a Ph.D. studentship from the BBSRC LiDO programme. We thank Professor David E. Clapham (Harvard Medical School, Boston, MA) for supplying the original NavMs gene and Dr Claire Bagneris (Birkbeck College) for providing the full-length construct. We thank Dr Ambrose Cole (Birkbeck College) for help with crystallographic data collection and the beamline scientists at the Diamond Light Source (I24), Soleil (Proxima 1 and 2), and European Synchrotron Radiation Facility (ID23) beamlines. The research leading to these results has received funding from the European Community's Seventh Framework Programme (FP7/2007–2013) under BioStruct-X (grant agreement N°283570).

## Author contributions

A.S. and J.B. cloned, expressed and purified the proteins, and J.B. created the I218C mutant. A.S. and J.B. collected the data, and A.S. and C.E.N. processed the data and solved the structures. A.S., J.B., C.E.N. and B.A.W. analysed the structures. P.G.D. supervised, designed, conducted and analysed the electrophysiology studies on the mutants with the help of L.C.T.N. All authors discussed the results. A.S. and J.B. created the graphics figures. B.A.W. initiated and supervised the project and wrote the initial manuscript, and all authors modified the manuscript and approved of the final version.

## Additional information

**Competing financial interests:** The authors declare no competing financial interests.

**Publisher's note**: 

