## [Peer Review File · Nature Communications]

Reviewer #2 (Remarks to the Author)

The manuscript by Sula and colleagues report a high resolution crystal structure for a complete prokaryotic voltage-gated sodium channel (NavM). The new structure is more complete than prior structures and includes the voltage-sensor domain (VSD), pore domain and the carboxyl-terminal domain (CTD). The structure has features consistent with an open state and there are novel interactions identified between the S4-S5 linker with a conserved residue (W77) in the S3 segment and additional interactions with a cluster of acidic residues within the extended part of the S6 helix (proximal CTD). These new findings definitely improve our understanding of Nav structure and raise new questions about the contribution of the CTD to gating phenomena. Although the manuscript has some important new observations, there are some potential shortcomings that should be considered and addressed.

Major concerns

1. The interactions between Trp-77 in S3 with the S4-S5 linker is particularly interesting and more work should be done to explore the functional relevance of this interaction. What happens functionally if this interaction is prevented say by mutating Trp-77 to alanine?
2. The presentation of the results is so intermingled with discussion points that the original data in the paper are difficult to distinguish from prior studies.
3. Exactly how does the interaction network involving S3, S4-S5 and proximal CTD explains impaired inactivation when CTD is mutated? Does inactivation of NavM involve these cytoplasmic facing structures? These interactions would appear most likely to stabilize the open state. It is not intuitively obvious why the authors state that this interaction network can rationalize the previously reported mutation findings.

Minor concern

a. State explicitly how you define the start of the CTD. In some parts of the text, you refer to the extended portion of S6 whereas elsewhere it is implied that this same region is the proximal part of the CTD. Be consistent with terms.

Reviewer #3 (Remarks to the Author)

The manuscript is coherent and the authors have responded clearly and positively to the comments of referee #1.

The crystallography (both data and refinement) is of excellent quality and the B statistics consistent with other ion channel structures. Annealed omit maps have now been provided, and show molecular detail in all of the requested regions. The electron density for both voltage sensor and the interaction domain connection is definitive. (One query re caption: possibly the right hand panels are 0.8 sigma and the left hand 1.1?).

Re Novelty: I concur with the authors and referee #2 that Sula et al makes novel findings and does not reiterate previous findings on other ion channels. Firstly, there are some key differences between K⁺, Na⁺ and Ca²⁺ channels. One of them is a motif discovered here. Secondly, no previous structure has clearly shown the contact regions - the detail in NavMs is unprecedented. Definition of the molecular connections between domains is essential to advance understanding of mechanism. In this instance, the new structures highlight the role of the C-terminal domain in gating.

Re "no major mechanistic insights": To the contrary, the TPC1 structure from Jiang's lab is refined to much lower resolution (3.3 Å) than NavMs (2.45 Å) and unlikely to have sufficient molecular detail to show specifics of the voltage sensor (no electron density close-ups in their paper). The Kv

chimera published by the Mackinnon lab (Long et al) was not well ordered in the vicinity of the intracellular face of the membrane. Importantly, the different cytoplasmic domains mean that neither of these distantly related structures have the same intramolecular contacts as NavMs. It would be nonsensical to patchwork their information with the structure of full-length KcsA, which is structurally distinct – KcsA lacks a voltage-sensor and has an unrelated Intracellular domain- and expect it to be informative. This issue was addressed directly in the author response.

Comparison of voltage sensors: It is important to show the different sets of pairings and Figure 2 is very informative. The apparently distinct ion pairings are not so different. The R3 interaction cluster in NavMs indicates D49, which interacts with both R2 and R3, is protonated (i.e. uncharged), inferring that hydrogen bonds (not ion pairs) exist - D49...R2 and D49...R3. Thus D49 is not behaving as a charged residue within the membrane, effectively similar to NaVAb. The R2 pairing partner E46 is not, I suspect, located within the membrane, but similarly placed to phospholipid moieties. In the activated state either could compensate for R2 charge. The transition at the lower R4 cluster is very revealing, and a first.

Comparison of S4-S5 linkers: The response and inclusion of the new overlay addresses all concerns.

The new paragraph (at end of rebuttal) seems warranted.

Point-by-Point Responses to the Reviewers
(Comments in italics, responses in bold)

Reviewer 2:

Paragraph 1: *The manuscript by Sula and colleagues report a high resolution crystal structure for a complete prokaryotic voltage-gated sodium channel (NavM). The new structure is more complete than prior structures and includes the voltage-sensor domain(VSD), pore domain and the carboxyl-terminal domain (CTD). The structure has features consistent with an open state and there are novel interactions identified between the S4-S5 linker with a conserved residue (W77) in the S3 segment and additional interactions with a cluster of acidic residues within the extended part of the S6 helix (proximal CTD). These new findings definitely improve our understanding of Nav structure and raise new questions about the contribution of the CTD to gating phenomena.*

Very supportive, no changes requested.

Paragraph 2: *Although the manuscript has some important new observations, there are some potential shortcomings that should be considered and addressed. Major concerns:*

They are addressed individually in response to the questions below.

1. *The interactions between Trp-77 in S3 with the S4-S5 linker is particularly interesting and more work should be done to explore the functional relevance of this interaction. What happens functionally if this interaction is prevented say by mutating Trp-77 to alanine?*

We agree this is a very interesting point, and to address the reviewer’s comment, we collaborated with Dr. Paul DeCaen (and his postdoc Leo Ng), who have undertaken the electrophysiology studies the Trp77 mutants, which are now described in the “Relationship of Structure to Function Properties” section of the Results and shown in new Figure 5.

Drs. DeCaen and Ng have been added as co-authors. The W77A mutation suggested by the reviewer resulted no functional current as did a W77M mutant), however less severe mutations (W77F and W77Y) produced channels with slower activation rates and positive shifts in the voltage dependence of activation and inactivation, as would be expected from the interaction domain structure described. We also made homology models of these mutants, as they did not crystallise, to illustrate the types of changes in the interaction domain these mutations could produce (below). They have been included as a new Extended Data Figure 8, for illustrative purposes:

2. *The presentation of the results is so intermingled with discussion points that the original data in the paper are difficult to distinguish from prior studies.*

We agree. The paper was originally written as a short letter to Nature, and was transferred to Nature Communications after a long review process. Because of this, the transferred version was very terse and not subdivided into introduction, results and conclusions sections. It has now been restructured (without changing the contents) to produce a normal article divided into sections and subsections, to address the reviewer's concern. We believe this significantly improves the paper.

3. *Exactly how does the interaction network involving S3, S4-S5 and proximal CTD explains impaired inactivation when CTD is mutated? Does inactivation of NavM involve these cytoplasmic facing structures? These interactions would appear most likely to stabilized the open state. It is not intuitively obvious why the authors state that this interaction network can rationalize the previously reported mutation findings.*

The reviewer is correct that this was not clear in the previous version. In the new subdivided version of the manuscript, and with the new functional data on the mutants, this has now been addressed in the "Relationship of Structure to Function" section.

Minor concern: State explicitly how you define the start of the CTD. In some parts of the text, you refer to the extended portion of S6 whereas elsewhere it is implied that this same region is the proximal part of the CTD. Be consistent with terms.

The reviewer is correct, and we have now defined it explicitly in the first paragraph of the subsection called "The C-terminal Domain and Novel Interaction Motif" in the results.

Reviewer 3:

Paragraph 1: The manuscript is coherent and the authors have responded clearly and positively to the comments of referee #1.

No changes required.

Paragraph 2: The crystallography (both data and refinement) is of excellent quality and the B statistics consistent with other ion channel structures. Annealed omit maps have now been provided, and show molecular detail in all of the requested regions. The electron density for both voltage sensor and the interaction domain connection is definitive.

No changes required.

(One query re caption: possibly the right hand panels are 0.8 sigma and the left hand 1.1?).

The reviewer is correct, the figure legends were reversed, and this has been corrected.

Paragraph 3: Re Novelty: I concur with the authors and referee #2 that Sula et al makes novel findings and does not reiterate previous findings on other ion channels. Firstly, there are some key differences between K⁺, Na⁺ and Ca²⁺ channels. One of them is a motif discovered here. Secondly, no previous structure has clearly shown the contact regions - the detail in NavMs is unprecedented. Definition of the molecular connections between domains is essential to advance understanding of mechanism. In this instance, the new structures highlight the role of the C-terminal domain in gating.

All comments are very supportive, and no changes required.

Paragraph 4: Re "no major mechanistic insights": To the contrary, the TPC1 structure from Jiang's lab is refined to much lower resolution (3.3 Å) than NavMs (2.45 Å) and unlikely to have sufficient molecular detail to show specifics of the voltage sensor (no electron density close-ups in their paper). The Kv chimera published by the Mackinnon lab (Long et al) was not well ordered in the vicinity of the intracellular face of the membrane. Importantly, the different cytoplasmic domains mean that neither of these distantly related structures have the same intramolecular contacts as NavMs. It would be nonsensical to patchwork their information with the structure of full-length KcsA, which is structurally distinct; KcsA lacks a voltage-sensor and has an unrelated Intracellular domain- and expect it to be informative. This issue was addressed directly in the author response.

No changes required.

Paragraph 5: Comparison of voltage sensors: It is important to show the different sets of pairings and Figure 2 is very informative.

Again, supportive and no changes required.

The apparently distinct ion pairings are not so different. The R3 interaction cluster in NavMs indicates D49, which interacts with both R2 and R3, is protonated (i.e. uncharged), inferring that hydrogen bonds (not ion pairs) exist - D49;R2 and D49;R3. Thus D49 is not behaving as a charged residue within the membrane, effectively similar to NaVAb. The R2 pairing partner E46 is not, I suspect, located within the membrane, but similarly placed to phospholipid moieties. In the activated state either could compensate for R2 charge.

Changes made to clarify as suggested.

The transition at the lower R4 cluster is very revealing, and a first.

This has now been noted in the text.

Paragraph 7: Comparison of S4-S5 linkers: The response and inclusion of the new overlay addresses all concerns.

No changes required.

Paragraph 8: The new paragraph (at end of rebuttal) seems warranted.

We have added it to the main text, as the reviewer has indicated.

In summary, other than reorganisation into the subsections required of a Nature Communications paper (which also clarifies the Results from the Conclusions), the only substantive question that either reviewer raised was requesting functional data to support the newly identified structural motif, which has now been included in the revised version.

Reviewer #2 (Remarks to the Author)

The revised manuscript by Sula and colleagues has been improved with changes that have addressed my prior concerns. There are a few minor points to consider:

1. Perhaps the 'Results' section would be more accurately labeled as 'Results and Discussion'.
2. In the new data figure 5b, the y-axis has a misspelled word: 'Denisity' should be 'Density'. Also in the legend for figure 5c, the 'rates' really should be called 'time constants'

Point-by-Point Response to REVIEWERS' COMMENTS: [reviewer's comments in italics; response in bold]

Reviewer #2 (Remarks to the Author):

The revised manuscript by Sula and colleagues has been improved with changes that have addressed my prior concerns. There are a few minor points to consider:

1. Perhaps the 'Results' section would be more accurately labeled as 'Results and Discussion'.

The notes from the journal provided with the editor's letter require that the Sections be called "Results" so we cannot do this, although we would be happy to do so.

2. In the new data figure 5b, the y-axis has a misspelled word: 'Denisity' should be 'Density'. Also in the legend for figure 5c, the 'rates' really should be called 'time constants'

These have been fixed.